# Magneto-Mechanical Enhancement of Elastic Moduli in Magnetoactive Elastomers with Anisotropic Microstructures

**DOI:** 10.3390/ma15020645

**Published:** 2022-01-15

**Authors:** Sanket Chougale, Dirk Romeis, Marina Saphiannikova

**Affiliations:** Leibniz Intitute of Polymer Research, 01069 Dresden, Germany; romeis@ipfdd.de (D.R.); grenzer@ipfdd.de (M.S.)

**Keywords:** magnetoactive elastomers, anisotropic particle distribution, smeared microstructures, effective elastic properties

## Abstract

Magnetoactive elastomers (MAEs) have gained significant attention in recent years due to their wide range of engineering applications. This paper investigates the important interplay between the particle microstructure and the sample shape of MAEs. A simple analytical expression is derived based on geometrical arguments to describe the particle distribution inside MAEs. In particular, smeared microstructures are considered instead of a discrete particle distribution. As a consequence of considering structured particle arrangements, the elastic free energy is anisotropic. It is formulated with the help of the rule of mixtures. We show that the enhancement of elastic moduli arises not only from the induced dipole–dipole interactions in the presence of an external magnetic field but also considerably from the change in the particle microstructure.

## 1. Introduction

Magnetizable particles embedded in soft elastomer matrix form a smart rubber composite known as magnetoactive elastomers (MAEs), whose mechanical and rheological properties can be manipulated externally with the magnetic field [1,2,3,4,5,6,7,8,9,10,11,12,13,14]. The ability of controlling mechanical properties externally with an applied magnetic field provides a promising technology for soft robotics and biomedical devices [15,16]. Thus, these MAEs can be used in a variety of engineering applications including but not limited to actuators, adaptive engine mounts, metamaterials, artificial cilia, retina magnetic fixators, tunable vibration absorbers, long-term biofilm control, etc. [17,18,19,20,21,22,23,24]. The fabrication of such MAEs under the application of an external magnetic field rearranges the randomly distributed magnetic particles into chain-like or plane-like microstructures [25,26,27,28,29,30]. If a homogeneous external magnetic field is applied during the cross-linking, particles tend to align into chains along the field direction. Alternatively, the use of a rotating magnetic field transforms the particle distribution to a plane-like microstructure [31]. Recently, anisotropic MAEs are also synthesized using novel 3D-printing techniques [32,33,34]. The mechanical properties of MAEs are highly sensitive to the initial shape of a sample [35,36,37,38] as well as to the particle microstructure [39,40,41]. MAEs possess the ability to change their elastic moduli in the presence of an external magnetic field [42,43,44,45]. To observe large enhancements in the moduli, one needs a very soft polymer matrix. Recently, ultra soft elastomeric matrices are introduced with the bottlebrush architecture [46]. A three-order increase in the shear modulus is demonstrated by using these “supersoft” elastomer matrices.

A variety of theoretical works can be found in the literature that investigate the effect of microstructure on the mechanical properties of MAEs [29,30,31,47,48,49,50,51,52]. In most works, the particle microstructure is described by the discrete particle positions inside an elastomer matrix. For that, different lattice models are considered. However, the precise particle positions are usually not known. Such lattice models show some pragmatic limitations due to the consideration of perfectly ordered microstructures. As an alternative, a different characterization scheme has been proposed in Ref. [53]. Instead of discrete particle distribution, the particle positions are smeared over an elongated columnar-like microstructure. The assumption of smeared microstructures replaces the discrete summation with an integral over the whole MAE sample. The transition from discrete summation to an integral significantly simplifies the model of MAEs with chain-like particle distribution. The present work extends this formalism to plane-like structures and attempts to simplify the formalism even further by converting the integral that describes the smeared particle microstructure in Ref. [53] into a simple analytical expression.

Such microstructures in MAEs also introduce a mechanical anisotropy in the material already in the absence of an external magnetic field [54,55]. Therefore, one has to consider an additional contribution due to anisotropic structures to the elastic free energy of MAEs. The anisotropic MAEs with smeared microstructures exhibit transverse isotropy along the symmetry axis, as illustrated in Figure 1. Transversely isotropic materials are also called unidirectional composites that show isotropic properties in the plane perpendicular to the preferred direction. Thus, the elastic free energy density is formulated by considering transverse isotropy in anisotropic MAEs. The dimensionless parameters related to the stretch of anisotropic microstructures are estimated using the rule of mixtures.

Following our previous works [37,38], an ellipsoidal MAE sample of two equal semi-axes and one distinct semi-axis is considered, as shown in Figure 2. We study the effect of different particle microstructures and the initial shape of an MAE sample on its mechanical properties. The magnetic particles are considered as point-like dipoles, and the linear magnetization regime is assumed. The paper is arranged as follows: In the next section, the material model of ellipsoidal MAE is presented. The simplification of the formalism presented in Ref. [53] is explained in detail by providing simple geometrical arguments. The magneto-induced deformations and magneto-rheological effects are investigated in Section 3 and Section 4, respectively. In the last section, conclusions are drawn, and the effect of particle rearrangement is discussed.

## 2. Materials and Methods

The deformation gradient tensor is defined as $F=\frac{\partial \vec{x}}{\partial \vec{X}}$, where $\vec{X}$ is the position vector of a material point in the reference configuration (undeformed), and $\vec{x}$ is the position vector in the current configuration (deformed). The resultant right and left Cauchy deformation tensors are $C=F^{T}\cdot F$, $b=F\cdot F^{T}$, respectively. The principal invariants of the right Cauchy deformation tensor are given as [56]
(1)$$I_{1}=tr\left(C\right),I_{2}=\frac{1}{2}\left({\left(tr\left(C\right)\right)}^{2}-tr\left(C^{2}\right)\right),I_{3}=det\left(C\right)=J^{2}$$
where *J* is the volume ratio between current and reference configurations [57]. For incompressible materials, J=1 [58]. It is a common practice to separate the elastic free energy of transversely isotropic materials into a contribution for the elasticity of the matrix material $\psi_{iso}$ and the influence of anisotropic structures $\psi_{aniso}$. The anisotropic part is defined with additional invariants, called “pseudo-invariants” $I_{4}$, $I_{5}$, and $I_{6}$, under rotations around the preferred direction in the material. They describe the effects of reinforcement due to the presence of rigid anisotropic structures [58,59].

The chain-like structures in MAEs can be approximated as fibers [54]. In the literature [56], the fiber-reinforced elastic composites are typically modeled as transversely isotropic materials. The pseudo invariant $I_{4}$ characterizes a family of fibers with some preferred direction (for instance, along unit vector ${\vec{a}}_{0}$). The invariant $I_{5}$ is only considered during the shear deformations, as shown in [59]. It is omitted for uniaxial elongation by considering a modification $I_{5}^{*}=I_{5}-I_{4}^{2}$. For uniaxial elongation, the invariant $I_{5}^{*}$ is always zero. Similarly, the MAEs with plane-like structures are also transversely isotropic materials [54]. However, in this case, it is not possible to model the plane-like structures with only pseudo invariant $I_{4}$, as we explain further in Section 2.2. In order to model such a microstructure, we consider additionally auxiliary pseudo invariant $I_{6}$ that describes the same family of fibers but with a different preferred direction (for example, along unit vector ${\vec{b}}_{0}$, and ${\vec{a}}_{0}∦{\vec{b}}_{0}$) [56].
(2)$$I_{4}={\vec{a}}_{0}\cdot C\cdot {\vec{a}}_{0},I_{5}={\vec{a}}_{0}\cdot C^{2}\cdot {\vec{a}}_{0},I_{6}={\vec{b}}_{0}\cdot C\cdot {\vec{b}}_{0}$$

In this work, we restrict ourselves to uniaxial deformations. Thus, we describe the total elastic free energy density in the following form:(3)$$\psi_{el}=\psi_{iso}\left(I_{1},I_{2},I_{3}\right)+\psi_{aniso}\left(I_{4},I_{6}\right).$$

Furthermore, we consider the isotropic elastic part as Neo-Hookean solid. The anisotropic elastic part can be described by a variety of forms reviewed in Ref. [59]. Anisotropic MAEs with chain-like structures can be approximated as elongated columnar structures or “fibers”. Such an alternative description for chain-like structures was introduced in Ref. [53]. In this work, we extend this description to plane-like structures. Since the precise particle positions are generally unknown, we smear them continuously over some expanded microstructure; see Figure 1. We approximate the chain-like structures as smeared columns (SCs) and plane-like structures as smeared disks (SDs). The anisotropic MAEs with SCs and SDs have different anisotropic contributions to the elastic free energy and magnetic energy densities.

In the linear magnetization regime, the total magnetic energy density in the sample is given by [31,37,53]: (4)$$\psi_{mag}=-\frac{\mu_{0}}{2V_{s}}\int_{V_{s}}d^{3}r\;\vec{M}\cdot {\vec{H}}_{0}$$
where $\mu_{0}=4\pi \times 10^{-7}$ NA${}^{-2}$ is the permeability of vacuum. The magnetization field $\vec{M}=\vec{M}\left(\vec{r}\right)$ depends on the sample shape and the particle distribution. We consider the homogeneous external magnetic field ${\vec{H}}_{0}$. Then, the volume integral in Equation (Equation 4) can be written as: (5)$$\psi_{mag}=-\frac{\mu_{0}}{2}\phi \langle \vec{M}\rangle \cdot {\vec{H}}_{0}$$
where $\langle \vec{M}\rangle$ denotes the average magnetization among all inclusions in the sample. Since the elastic matrix is not magnetizable, the factor ϕ, the total volume fraction of particles, enters Equation (Equation 5).

Previous works showed that assuming an ellipsoidal sample shape, the magnetic energy can be decomposed into two independent contributions. One represents the macroscopic shape of the sample, $f_{macro}$, and the other refers to the microscopic particle distribution, $f_{micro}$ [31,53]. Here, $f_{macro}$ is closely related to the demagnetizing factor of a homogeneously magnetized ellipsoid. In order to model the microscopic contribution $f_{micro}$, the individual particle positions must be presumed. For example, this can be achieved considering lattice-like particle distributions [29,31,47]. However, as mentioned previously, the precise particle positions are unknown in realistic samples [53]. Furthermore, we consider constant density $\phi_{p}$ within smeared structures, as depicted in Figure 1. Thus, we define $\phi_{p}$ as the volume fraction of particles inside a smeared structure, and it follows: $\phi_{p}\geq \phi$. The volume fraction of smeared structures (at $\phi_{p}>\phi$) is obtained as the ratio of $\phi_{f}=\frac{\phi }{\phi_{p}}$ [54]. At $\phi_{p}=\phi$, the particle density is the same all over the sample describing the isotropic distribution of particles (no smeared structures) and consequently $\phi_{f}=1$. To attain magnetic energy for samples with SCs, the locally varying magnetization field $\vec{M}\left(\vec{r}\right)$ was calculated self-consistently in Ref. [53]. Such “full” self-consistent treatment requires an elaborate formulation, and the solution can be computed only numerically.

Recently, an efficient approximation scheme [41,60] could be established to calculate magnetization fields in composite materials under rather general conditions. For example, a tensorial notation was introduced to describe the effects of arbitrarily oriented external magnetic fields and/or more generic particle microstructures or sample shapes. In particular, the tensor $G_{\mathrm{micro}}$ was introduced to describe the microstructure. In the present work, we consider the external magnetic field ${\vec{H}}_{0}$ aligned with the symmetry axis of the particle structure. Furthermore, also the symmetry axis of the sample form itself, i.e., ellipsoid of revolution, is co-aligned with ${\vec{H}}_{0}$. Accordingly, the tensorial notation can be reduced to a scalar description with all fields oriented along the *x*-direction, i.e., ${\vec{H}}_{0}=H_{0}\vec{e_{1}}$, $\vec{M}=M\vec{e_{1}}$ and consequently also the total magnetic field $\vec{H}=H\vec{e_{1}}$ (a unit vector ${\vec{e}}_{1}$ is aligned along the *x*-direction). Assuming the linear magnetization regime, $\vec{M}=\chi \vec{H}$ with isotropic susceptibility χ, the average magnetization in the sample is found via the leading order approximation [41,60] as:(6)$$\langle M\rangle =\frac{\chi_{eff}H_{0}}{1-\chi_{eff}\left(\phi f_{macro}+f_{micro}\right)}.$$

Here, $\chi_{eff}=\frac{\chi }{1+\chi n_{d}}$ denotes the effective susceptibility, with $n_{d}$ being the particle demagnetization factor. Considering spherical inclusions, we have $n_{d}=\frac{1}{3}$. The macroscopic contribution from the sample shape for ellipsoidal MAEs reads $f_{macro}=\frac{1}{3}-J_{a}$, where $J_{a}$ is the demagnetizing factor of an ellipsoid along its symmetry axis [37]. According to Refs. [53,60], the contribution due to microscopic particle structure is formally obtained as:(7)$$f_{micro}=\;{\left(G_{micro}\right)}_{11}\;=\;\frac{1}{4\pi V_{MS}}{\;\int \int \;}_{V_{MS}}\;\;\;\;d^{3}r\;\;d^{3}r^{\prime }\;\Phi \left(\vec{r^{\prime }}\right)\frac{3{\left(x^{\prime }\;-\;x\right)}^{2}-{|\vec{r^{\prime }}\;-\;\vec{r}|}^{2}}{{|\vec{r^{\prime }}-\vec{r}|}^{5}}\Theta \left(|\vec{r^{\prime }}\;-\;\vec{r}|-d_{p}\right).$$

Here, $V_{MS}$ denotes a mesoscopic portion of the sample where the local particle structure is resolved, i.e., a representative volume in form of a mesoscopic sphere [53,60]. Since in the present approach, we describe the particle distribution as continuous (locally varying) fields, Equation (Equation 7) is formulated in terms of integrals instead of discrete summations over explicit particle positions. The particles are assumed to be of spherical shape with diameter $d_{p}$ and, accordingly, the Heavyside step function Θ(x) is introduced to restrict the integration to positions outside of the particle located at $\vec{r}$ (no self-interaction of particle positions) [53,60]. Thus, from Equations (Equation 5)–(Equation 7), the magnetic energy density is
(8)$$\psi_{mag}=-\frac{\mu_{0}\phi H_{0}^{2}}{2}\left(\frac{1}{R+\phi J_{a}-f_{micro}}\right)$$
where $R=\chi^{-1}+\frac{1}{3}-\frac{\phi }{3}$ [38]. The total free energy density of anisotropic MAEs is now a combination of three contributions: $\psi_{iso}$, $\psi_{aniso}$, and $\psi_{mag}$. As mentioned previously, by considering the isotropic elastic part as Neo-Hookean solid, the total free energy density can be given as
(9)$$\psi_{MAE}=\frac{G_{iso}}{2}\left(I_{1}-3\right)+\psi_{aniso}\left(I_{4},I_{6}\right)-\frac{\mu_{0}\phi H_{0}^{2}}{2}\left(\frac{1}{R+\phi J_{a}-f_{micro}}\right).$$

Here, $G_{iso}=G_{m}k_{iso}$ is the effective shear modulus of an isotropic MAE, and $k_{iso}$ is the hydrodynamic reinforcement factor [61,62] obtained via the rule of mixtures (see Appendix A).
(10)$$k_{iso}=\frac{G_{iso}}{G_{m}}=\left(1+\frac{2.5\phi }{1-2\phi }\right)$$
where $G_{m}$ is the shear modulus of a pure elastomer matrix. Equation (Equation 9) represents a general form of the free energy density of anisotropic MAEs. One needs to choose the appropriate form of $\psi_{aniso}$ depending on the microstructure under consideration. The values of the dimensionless parameter $f_{micro}$ also change with respect to the particle distribution. In the following section, we derive the specific form of $\psi_{aniso}$ and $f_{micro}$ for MAEs with SCs and SDs.

### 2.1. Free Energy of Anisotropic MAEs with Smeared Columns

Anisotropic MAEs with SCs can be approximated as fiber-reinforced materials that exhibit unidirectional anisotropy along the fibers. For such materials, we consider a quadratic form of $I_{4}$ as given in Ref. [59].
(11)$$\psi_{aniso}=\frac{G_{iso}}{2}\zeta_{SC}{\left(I_{4}-1\right)}^{2}$$
where the dimensionless parameter $\zeta_{SC}$ describes the fiber (smeared column) stretch. In this case, $I_{4}$ is invariant under the rotations around a unit vector ${\vec{e}}_{1}\equiv \left(1,0,0\right)$, as shown in Figure 1. Thus,
(12)$$I_{4}={\vec{e}}_{1}\cdot C\cdot {\vec{e}}_{1}=C_{11}.$$

The SCs are all aligned in the same direction, and thus, they are described by only one pseudo-invariant $I_{4}$ (thus, here, $I_{6}=0$). The longitudinal elastic modulus $E_{L}$ of MAEs with SCs along the symmetry axis is larger than the elastic modulus $E_{T}$ in the transverse direction. We compare the longitudinal elastic modulus derived from the elastic free energy of anisotropic MAEs having SCs with the modulus predicted from the rule of the mixtures to obtain values of the dimensionless parameter $\zeta_{SC}$; for details, see Appendix B. For MAEs with SCs,
(13)$$\zeta_{SC}=\frac{3}{4}\left(\frac{1-\phi_{f}+k\phi_{f}}{k_{iso}}-1\right)$$
where *k* is also the hydrodynamic reinforcement factor [61,62] given as
(14)$$k=\frac{G_{f}}{G_{m}}=1+\frac{2.5\phi_{p}}{1-2\phi_{p}}$$
where $G_{f}$ is the shear modulus of the isotropic fiber/smeared column. In the present formulation, an isotropic distribution is realized when $\phi_{p}=\phi$, and consequently $\phi_{f}=1$. Thus, one obtains following relations:(15)$$k=k_{iso},\;G_{f}=G_{iso},\;\zeta_{SC}=0.$$

In Ref. [53], the form of $f_{micro}$ for columnar structures has been studied in detail. There, a self-consistent treatment with locally varying magnetization $\vec{M}\left(\vec{r}\right)$ within the microstructure is derived. Here, we aim to provide an approximate, but in return, analytic form for the microstructure effect. In the following, we make use of some relations provided in Ref. [53]. We note that in smeared structures along ${\vec{e}}_{1}$, the local particle volume fraction $\Phi \left(\vec{r}\right)$ does not depend on the *x*-coordinate $\left(\Phi \neq \Phi \left(x\right)\right)$. Then, the contributions to $f_{micro}$ originating from material portions situated at finite lateral distances (y−z-directions) with respect to the reference location, i.e., position $\vec{r}$ in Equation (Equation 7), vanish. A non-zero contribution results from particles found above and below the reference particle *i*, see Figure 2, and we denote it as $f_{micro}^{\prime }$. Another non-zero contribution relates to volume portions located sufficiently far away so that the micro-structure is not resolvable anymore and the particle distribution appears homogeneous with $\Phi \left(\vec{r}\right)=\phi$. The corresponding share to $f_{micro}$ evaluates to $-\frac{\phi }{3}$ [53].

In order to calculate the contribution $f_{micro}^{\prime }$, we neglect effects due to particles located exactly on or close to the boundaries of a smeared column. Neglecting such ‘boundary’ effects has the beneficial outcome that $f_{micro}^{\prime }$, and thus $f_{micro}$ altogether, adopts a very simple analytic form. Accounting for the boundaries of particle-containing columns results in an explicit dependency on the actual lateral size, or diameter, of the columns. Upon introducing the elastic free energy, we describe the mechanical effect of particle microstructures in terms of fiber-like structures with enhanced stiffness parameter $\zeta_{SC}$. The formulation is restricted to the parameters $\phi_{p}$ and $\phi_{f}$. No dependency on the thickness of the smeared columns is presumed. Accordingly, neglecting such structural size effects in the magnetic formulation represents a consistent simplification. Assuming any reference particle positioned well inside the columnar structure, and considering the particle volume fraction in such column as constant with $\Phi =\phi_{p}$, we note that every particle experiences an identical filler concentration above and below its actual position. Consequently, the contribution $f_{micro}^{\prime }$ is calculated as [53]
(16)$$f_{micro}^{\prime }=\phi_{p}\int_{\sqrt{d_{p}^{2}-\rho^{2}}}^{\infty }dx\int_{0}^{d_{p}}\rho d\rho \frac{2x^{2}-\rho^{2}}{{\left(x^{2}+\rho^{2}\right)}^{5/2}}=\frac{\phi_{p}}{3}$$
and the total ${\left(f_{micro}\right)}_{SC}$ in the case of SCs reads:(17)$${\left(f_{micro}\right)}_{SC}=\frac{1}{3}\left(\phi_{p}-\phi \right).$$

This expression is remarkably neat and compact. Note that Equation (Equation 17) correctly reproduces the result for an isotropic particle distribution, i.e., $f_{micro}=0$ at $\phi_{p}=\phi$ [31,41,63]. The formation of columnar structures requires $\phi_{p}>\phi$ and in turn, $f_{micro}>0$ [31,41,63]. By substituting Equations (Equation 11) and (Equation 17) into (Equation 9), we obtain
(18)$$\psi_{MAE}=\frac{G_{iso}}{2}\left(\left(I_{1}-3\right)+\zeta_{SC}{\left(I_{4}-1\right)}^{2}-\frac{\mu_{0}\phi H_{0}^{2}}{2G_{iso}}\left(\frac{1}{R+\phi J_{a}-{\left(f_{micro}\right)}_{SC}}\right)\right).$$

Equation (Equation 18) refers to the specific form of the free energy density of anisotropic MAEs with SCs.

### 2.2. Free Energy of Anisotropic MAEs with Smeared Disks

We consider the plane-like microstructure of MAEs as smeared disks, as shown in Figure 1. In this case, too, MAEs exhibit transverse isotropy along the symmetry axis of SDs. However, SDs require at least two invariants to describe the plane of isotropy, which is perpendicular to a unit vector ${\vec{e}}_{1}$. Thus, here, we consider two pseudo invariants $I_{4}$ and $I_{6}$ to take into account the anisotropic contribution due to SDs to the elastic free energy of MAEs. We define $I_{4}$ and $I_{6}$ with respect to unit vectors ${\vec{e}}_{2}\equiv \left(0,1,0\right)$ and ${\vec{e}}_{3}\equiv \left(0,0,1\right)$ as
(19)$$\begin{array}{c} I_{4}={\vec{e}}_{2}\cdot C\cdot {\vec{e}}_{2}=C_{22}\\ I_{6}={\vec{e}}_{3}\cdot C\cdot {\vec{e}}_{3}=C_{33} \end{array}.$$

The unit vectors ${\vec{e}}_{2}$ and ${\vec{e}}_{3}$ are perpendicular to each other and also to the direction of anisotropy ${\vec{e}}_{1}$ such that ${\vec{e}}_{2}\cdot {\vec{e}}_{3}=0$ and ${\vec{e}}_{1}={\vec{e}}_{2}\times {\vec{e}}_{3}$, as shown in Figure 1. The invariants $I_{4}$ and $I_{6}$ are typically used in the modeling of transversely isotropic materials with two families of fibers. In this work, we consider a disk (or plane) formed by a single family of fibers but with directions along ${\vec{e}}_{2}$ and ${\vec{e}}_{3}$, retaining the preferred direction the same as in the previous case (along ${\vec{e}}_{1}$). As the SD has uniform properties around its symmetry axis, the mathematical manipulation of considering two directions such that ${\vec{e}}_{2}\cdot {\vec{e}}_{3}=0$ does not lead to orthotropic materials [56], keeping the material transversely isotropic. By considering the quadratic form of $I_{4}$ and $I_{6}$, we propose the following anisotropic contribution of SDs
(20)$$\psi_{aniso}=\frac{G_{iso}}{2}\zeta_{SD}\left({\left(I_{4}-1\right)}^{2}+{\left(I_{6}-1\right)}^{2}\right)$$
where $\zeta_{SD}$ is related to the stretch of SDs in anisotropic MAEs. Unlike in the previous sections, MAEs with SDs have larger transverse modulus $E_{T}$ in the plane perpendicular the symmetry axis (${\vec{e}}_{1}$) and smaller $E_{L}$ along this axis. In this case, we compare the transverse modulus calculated from the free energy of MAEs with SDs using Equation (Equation 20) to the modulus obtained from the rule of mixtures to estimate the value of $\zeta_{SD}$ (see Appendix C for more details). Here, an analytical expression for $\zeta_{SD}$ is not possible, and the solution is calculated numerically. With the proposed $\psi_{aniso}$ for MAEs with SDs, one can attain total elastic free energy density. Analogous to Equation (Equation 15), we have
(21)$$\zeta_{SD}{|}_{\phi_{p}=\phi }=0.$$

The contribution of SDs to magnetic energy density also differs from the previous case of MAEs with SCs. The prefactor $\frac{1}{3}$ for SCs in Equation (Equation 17) can be easily understood from geometrical considerations. Smeared columns may be interpreted as infinitely long cylinders or as prolate spheroids with an infinitely large aspect ratio γ. The demagnetizing factor along such spheroid reads $J_{a}\left(\gamma \to \infty \right)=0$, and the shape factor becomes $f=\frac{1}{3}$. Analogously, we may describe smeared disks as infinitely expanded oblate spheroids with vanishing γ. The demagnetizing factor for such objects turns to $J_{a}\left(\gamma \to 0\right)=1$, and consequently, we immediately find:(22)$${\left(f_{micro}\right)}_{SD}=-\frac{2}{3}\left(\phi_{p}-\phi \right).$$

Equivalently, this result may be derived from the explicit calculation of Equation (Equation 7). By combining the Equations (Equation 9), (Equation 20) and (Equation 22), the total free energy density of an anisotropic MAE with SDs reads: (23)$$\psi_{MAE}\;=\;\frac{G_{iso}}{2}\left(\left(I_{1}\;-\;3\right)\;+\;\zeta_{SD}\left({\left(I_{4}\;-\;1\right)}^{2}\;+\;{\left(I_{6}\;-\;1\right)}^{2}\right)\;-\;\frac{\mu_{0}\phi H_{0}^{2}}{2G_{iso}}\left(\frac{1}{R\;+\;\phi J_{a}\;-\;{\left(f_{micro}\right)}_{SD}}\right)\right).$$

As mentioned earlier, for $\phi_{p}=\phi$, one has the isotropic particle distribution and subsequently, $\zeta_{SC}=\zeta_{SD}={\left(f_{micro}\right)}_{SC}={\left(f_{micro}\right)}_{SD}=0$. Thus, from Equations (Equation 15) and (Equation 21), we have $\psi_{el}=\psi_{iso}$. Therefore, the present formalism is fully consistent with the previous studies of isotropic MAEs [31,37,38,53].

## 3. Magneto-Induced Deformations

The tensile mechanical test is a destructive process that characterizes the tensile strength and the extent to which the sample elongates [64]. In the case of MAE, the tensile tests are carried out in the presence of an external magnetic field ${\vec{H}}_{0}$. This section investigates the magneto-induced elongation of anisotropic MAEs for different volume fractions of magnetic particles. The unit vector ${\vec{e}}_{1}$ and applied magnetic field ${\vec{H}}_{0}$ are aligned along the *x*-axis, as shown in Figure 3. The uniaxial deformation gradient tensor in matrix form can be given as
(24)$$F=\left[\begin{array}{ccc} \lambda_{1} & 0 & 0\\ 0 & \lambda_{2} & 0\\ 0 & 0 & \lambda_{3} \end{array}\right]$$
where $\lambda_{1},\;\lambda_{2},\;$ and $\;\lambda_{3}$ are the stretch ratios along the x,y, and *z*-directions, respectively. The incompressibility condition states:(25)$$\lambda_{1}\lambda_{2}\lambda_{3}=1.$$

The demagnetizing factor of an ellipsoid $J_{a}$ along its symmetry axis is a function of aspect ratios $\gamma_{1}$ and $\gamma_{2}$. The change in aspect ratio is governed by the applied loading (mechanical or magnetic loadings) as:(26)$$\gamma_{1}=\gamma_{0}\frac{\lambda_{1}}{\lambda_{2}},\gamma_{2}=\gamma_{0}\frac{\lambda_{1}}{\lambda_{3}}$$
where $\gamma_{0}=\frac{a_{0}}{b_{0}}=\frac{a_{0}}{c_{0}}$ is the initial aspect ratio of a spheroidal MAE sample; see Figure 2. Accordingly, the demagnetizing factor $J_{a}$ is a function of the deformation gradient tensor *F*. We choose the value of magnetic susceptibility χ=1000 to model highly magnetizable material such as carbonyl iron with χ≫1 [10,31,53]. As the linear magnetization regime is assumed, we restrict the magnitude of applied magnetic field to a maximum value of 470 kA/m. A very soft elastomer matrix of shear modulus $G_{m}=17$ kPa is considered [46] to achieve maximum field-induced effects. All these parameters are summarized in Table 1.

With the deformation gradient tensor **F** and Equation (Equation 9), the Cauchy stress tensor of the MAE in the general case reads: (27)$$\sigma_{\mathrm{MAE}}=-pI+G_{iso}b+\frac{\partial \psi_{ansio}}{\partial F}F^{T}+G_{iso}\eta f_{N}\frac{\partial J_{a}}{\partial F}F^{T}.$$

In the above Equation (Equation 27), we introduced two dimensionless parameters:(28)$$\eta =\frac{\mu_{0}\phi^{2}{H_{0}}^{2}}{2G_{iso}}$$
and
(29)$$f_{N}=\frac{1}{{\left(R+\phi J_{a}-f_{micro}\right)}^{2}}.$$

Note that $f_{micro}$ does not depend on the actual size of a smeared structure. Thus, the microstructure deformation is neglected ($\frac{\partial f_{micro}}{\partial F}=0$). In the following sections, we investigate the magneto-induced elongations and magneto-rheological effects of MAEs with SCs and SDs. Accordingly, we substitute the expressions of $\psi_{aniso}$ and $f_{micro}$ in Equation (Equation 27).

### 3.1. Smeared Columns

Here, we examine the uniaxial elongation of anisotropic MAEs with SCs under the application of an external magnetic field, as shown in Figure 3. We apply the external magnetic field along the symmetry axis of a spheroidal MAE sample of the initial aspect ratio $\gamma_{0}$ and calculate the magneto-induced elongation in the applied field direction. For SCs, the values of dimensionless parameters ${\left(f_{micro}\right)}_{SC}$ and ${\left(\zeta \right)}_{SC}$ are positive and given by Equations (Equation 13) and (Equation 17), respectively. By substituting Equations (Equation 11) and (Equation 17) into (Equation 27), the corresponding Cauchy stress components can be calculated as:(30)$$\begin{array}{c} \begin{array}{cc} \sigma_{11} & =-p+G_{iso}\left(\left(\lambda_{1}^{2}+2\zeta_{SC}\left(\lambda_{1}^{2}-1\right)\lambda_{1}^{2}\right)+\eta f_{N}\left(J_{1}^{\prime }+J_{2}^{\prime }\right)\right)\\ \sigma_{22} & =-p+G_{iso}\left(\lambda_{2}^{2}-\eta f_{N}J_{1}^{\prime }\right)\\ \sigma_{33} & =-p+G_{iso}\left(\lambda_{3}^{2}-\eta f_{N}J_{2}^{\prime }\right) \end{array} \end{array}$$
where $J_{1}^{\prime }=\frac{\partial J_{a}}{\partial \gamma_{1}}\gamma_{1}$ and $J_{2}^{\prime }=\frac{\partial J_{a}}{\partial \gamma_{2}}\gamma_{2}$. To calculate the magneto-induced elongation, we consider $\sigma_{22}=\sigma_{33}=0$. From incompressibility condition (Equation 25) and Equation (Equation 30), we receive the relationship between the stretch ratios and the hydrostatic pressure *p*
(31)$$\begin{array}{c} \lambda_{2}=\frac{1}{\sqrt{\lambda_{1}}}\\ p=\frac{G_{iso}}{\lambda_{1}} \end{array}.$$

Thus, the non-zero Cauchy stress component along the symmetry axis of an MAE sample is
(32)$${\left(\sigma_{11}\right)}_{SC}=G_{iso}\left(\lambda_{1}^{2}-\frac{1}{\lambda_{1}}+2\zeta_{SC}\left(\lambda_{1}^{2}-1\right)\lambda_{1}^{2}\right)+\eta f_{N}\left(2J_{1}^{\prime }+J_{2}^{\prime }\right).$$

The stretch ratio $\lambda_{1}=\lambda_{1}^{H}\lambda_{1}^{m}$ in Equation (Equation 32) is a total stretch [37] combining: (1) a stretch due to the applied magnetic field $\lambda_{1}^{H}$ and (2) a stretch due to external mechanical loadings $\lambda_{1}^{m}$. In this section, we consider purely magnetic loadings. Hence, $\lambda_{1}^{m}=1$. The magneto-induced (equilibrium) elongation $\left(\lambda_{1}^{H}=\lambda_{eq}\right)$ is calculated at equilibrium condition when ${\left(\sigma_{11}\right)}_{SC}=0$. Figure 4 shows the magneto-induced elongation of an MAE with SC microstructure as a function of the initial aspect ratio $\gamma_{0}$ and the volume fraction of particles inside a smeared column $\phi_{p}$ at different total volume fractions ϕ. The equilibrium elongation $\lambda_{eq}$ decreases with an increase in $\phi_{p}$ at constant total particle volume fraction ϕ. The volume fraction inside an elongated column $\phi_{p}$ is directly related to the column’s strength. Consequently, the dimensionless parameter $\zeta_{SC}$ is strongly increasing as $\phi_{p}\to 0.5$. Furthermore, the strengthening of columns, especially when $\phi_{p}\to 0.5$, outweighs the magnetic field effect. As a result, magneto-induced elongation reduces. The optimal initial aspect ratio $\gamma_{0}$, where the maximum magneto-induced elongation is predicted, shifts toward higher values with an increase in $\phi_{p}$ and an overall decrease in the magnitude of $\lambda_{eq}$ (for example, refer to Appendix D).

### 3.2. Smeared Disks

For MAEs with SDs, the value of the dimensionless parameter ${\left(f_{micro}\right)}_{SD}$ is negative while $\zeta_{SD}$ is positive. The Cauchy stress components for MAEs with SDs are derived from Equations (Equation 20), (Equation 22) and (Equation 27) as
(33)$$\begin{array}{c} \begin{array}{cc} \sigma_{11} & =-p+G_{iso}\left(\lambda_{1}^{2}+\eta f_{N}\left(J_{1}^{\prime }+J_{2}^{\prime }\right)\right)\\ \sigma_{22} & =-p+G_{iso}\left(\lambda_{2}^{2}+2\zeta_{SD}\left(\lambda_{2}^{2}-1\right)\lambda_{2}^{2}-\eta f_{N}J_{1}^{\prime }\right)\\ \sigma_{33} & =-p+G_{iso}\left(\lambda_{3}^{2}+2\zeta_{SD}\left(\lambda_{3}^{2}-1\right)\lambda_{3}^{2}-\eta f_{N}J_{2}^{\prime }\right) \end{array}. \end{array}$$

Similar to the previous case, here, $\sigma_{22}=\sigma_{33}=0$, and the relationship between stretch ratios and the hydrostatic pressure *p* is given by Equation (Equation 31). Thus, the non-zero Cauchy stress component along the field direction for MAEs with SDs is
(34)$${\left(\sigma_{11}\right)}_{SD}=G_{iso}\left(\lambda_{1}^{2}-\left(\frac{1}{\lambda_{1}}+2\zeta_{SD}\left(\frac{1}{\lambda_{1}}-1\right)\frac{1}{\lambda_{1}}\right)+\eta f_{N}\left(2J_{1}^{\prime }+J_{2}^{\prime }\right)\right).$$

The effect of $\phi_{p}$ and ϕ on the magneto-induced elongation of MAEs with SD structures is investigated as a function of the initial aspect ratio $\gamma_{0}$. Analogous to SC structures, the equilibrium elongation $\lambda_{eq}$ decreases with an increase in the values of $\phi_{p}$ at constant ϕ, as seen in Figure 5. Here also, as $\phi_{p}\to 0.5$, the dimensionless parameter $\zeta_{SD}$ becomes very high, which results in the overall decrease in the magneto-induced elongations. In contrast to the previous section, in this case, the shifting of maxima is negligible, as illustrated in the Appendix D. Both SCs and SDs lead to an overall elongation of an MAE sample. It is because the surrounding columns and disks do not interact, and the shape effect is dominant. In addition, the elongation $\lambda_{eq}$ is more pronounced for SCs ($f_{micro}>0$) than SDs ($f_{micro}<0$) because the magnetic energy increases with an increase in $f_{micro}$; see Equation (Equation 8). Nevertheless, when $\phi_{p}\to 0.5$, the elastic parameters $\zeta_{SC}$ and $\zeta_{SD}$ dominate the effect of the applied magnetic field.

## 4. Magneto-Rheological Effect

The magneto-rheological effect (MR) is defined as the change in the elastic moduli of the MAE in the presence of an external magnetic field [37]. The initial shape of an MAE sample affects the MR effect significantly, as already shown in our previous works [37,38]. Here, along with the initial shape, we also study the effect of different microstructures (SCs and SDs).

### 4.1. Smeared Columns

The longitudinal elastic modulus $E_{\|}$ of an ellipsoidal MAE sample with SCs in the presence of an external magnetic field is calculated by taking the derivative of the Cauchy stress component ${\left(\sigma_{11}\right)}_{SC}$ over the stretch ratio $\lambda_{1}$ at $\lambda_{1}=\lambda_{eq}$. Similarly, the transverse Cauchy stress component $\sigma_{22}$ is needed to calculate the MR effect perpendicular to the field direction. For that, we consider a uniaxial elongation applied perpendicular to the field direction. Thus, in this case, $\sigma_{11}=\sigma_{33}=0$. From the incompressibility condition (Equation 25) and Equations (Equation 30), the stretch relation is given as:(35)$$\lambda_{2}=\frac{1}{\lambda_{1}\left(\lambda_{1}^{2}+2\zeta_{SC}\left(\lambda_{1}^{2}-1\right)\lambda_{1}^{2}\right)+\eta f_{N}{\left(J_{1}^{\prime }+2J_{2}^{\prime }\right)}^{1/2}}.$$

Here, $\lambda_{2}=\lambda_{2}^{H}\lambda_{2}^{m}$, where $\lambda_{2}^{H}=\frac{1}{\sqrt{\lambda_{eq}}}$. Considering the stretch relation in Equation (Equation 35), we derive the transverse Cauchy stress component
(36)$${\left(\sigma_{22}\right)}_{SC}=G_{iso}\left(\left(\lambda_{2}^{2}-\left(\lambda_{1}^{2}+2\zeta_{SC}\left(\lambda_{1}^{2}-1\right)\lambda_{1}^{2}\right)\right)-\eta f_{N}\left(2J_{1}^{\prime }+J_{2}^{\prime }\right)\right).$$

The expression for transverse elastic modulus ($E_{⊥}$) is obtained by taking the derivative of Equation (Equation 36) over $\lambda_{2}^{m}$. The elastic modulus ($E_{⊥}$) is calculated at $\lambda_{2}^{m}=1$ and $\lambda_{2}^{H}=\frac{1}{\sqrt{\lambda_{eq}}}$. Thus, the elastic moduli (at $|{\vec{H}}_{0}|\neq 0$) of MAEs with SC microstructure are given as:(37)$$\begin{array}{c} {\left(E_{\|}\right)}_{SC}=\frac{\partial {\left(\sigma_{11}\right)}_{SC}}{\partial \lambda_{1}}{|}_{H_{0}\neq 0}\\ {\left(E_{⊥}\right)}_{SC}=\frac{\partial {\left(\sigma_{22}\right)}_{SC}}{\partial \lambda_{2}}{|}_{H_{0}\neq 0} \end{array}.$$

The % MR-effect of MAEs with SCs is calculated as
(38)$$\begin{array}{c} {\left(K_{\|}\right)}_{SC}=\frac{{\left(E_{\|}\right)}_{SC}{|}_{H_{0}\neq 0}-{\left(E_{\|}\right)}_{SC}{|}_{H_{0}=0}}{{\left(E_{\|}\right)}_{SC}{|}_{H_{0}=0}}\times 100\\ {\left(K_{⊥}\right)}_{SC}=\frac{{\left(E_{⊥}\right)}_{SC}{|}_{H_{0}\neq 0}-{\left(E_{⊥}\right)}_{SC}{|}_{H_{0}=0}}{{\left(E_{⊥}\right)}_{SC}{|}_{H_{0}=0}}\times 100 \end{array}.$$

Similar to isotropic MAEs [37], the MR effect of anisotropic MAEs with SC microstructure is positive along the field direction and is negative transverse to it, as illustrated in Figure 6 and Figure 7. The MR effect along the field direction for SC microstructure ${\left(K_{\|}\right)}_{SC}$ increases with the volume fraction of particles inside a smeared structure $\phi_{p}$ for different total volume fractions (ϕ=0.15, ϕ=0.2, ϕ=0.3). However, after a critical value of $\phi_{p}\approx 0.4$, irrespective of the total volume fraction ϕ, the effects begin to vanish, as seen in Figure 6. It is the consequence of an increase in the effective elastic modulus of an elastomer composite due to higher values of $\zeta_{SC}$ when $\phi_{p}\geq 0.4$.

For the MR effect perpendicular to ${\vec{H}}_{0}$, the magnitude of ${\left(K_{⊥}\right)}_{SC}$ increases with an increase in $\phi_{p}$ for low total volume fractions ϕ=0.15, ϕ=0.2 (see Figure 7A,B), and it decreases for ϕ=0.3, as shown in Figure 7C.

### 4.2. Smeared Disks

In this case, to calculate the longitudinal elastic modulus of MAEs with SDs, we use ${\left(\sigma_{11}\right)}_{SD}$. Analogous to the previous section, to calculate the transverse Cauchy stress component ${\left(\sigma_{22}\right)}_{SD}$, we consider a uniaxial elongation applied perpendicular to the field direction. Thus, as explained previously, $\sigma_{11}=\sigma_{33}=0$. From the incompressibility condition (Equation 25) and Equation (Equation 33), the stretch relation for MAEs with SDs is given as:(39)$$\lambda_{2}=\frac{1}{\lambda_{3}\left(\lambda_{3}^{2}+2\zeta_{SD}\left(\lambda_{3}^{2}-1\right)\lambda_{3}^{2}\right)-\eta f_{N}{\left(J_{1}^{\prime }+2J_{2}^{\prime }\right)}^{1/2}}$$
where $\lambda_{3}=\frac{1}{\lambda_{1}\lambda_{2}}$. Note that again $\lambda_{2}=\lambda_{2}^{H}\lambda_{2}^{m}$, and $\lambda_{2}^{H}=\frac{1}{\sqrt{\lambda_{eq}}}$. Thus, the transverse Cauchy stress component is
(40)$${\left(\sigma_{22}\right)}_{SD}=G_{iso}\left(\lambda_{2}^{2}+2\zeta_{SD}\left(\lambda_{2}^{2}-1\right)\lambda_{2}^{2}-\lambda_{1}^{2}-\eta f_{N}\left(2J_{1}^{\prime }+J_{2}^{\prime }\right)\right).$$

By substituting the Cauchy stress components ${\left(\sigma_{11}\right)}_{SD}$ and ${\left(\sigma_{22}\right)}_{SD}$ in Equation (Equation 37), we obtain the elastic moduli and consequently the relative MR effects ${\left(K_{\|}\right)}_{SD}$ and ${\left(K_{⊥}\right)}_{SD}$ of MAEs with SDs.

Contrary to MAEs with SCs, the magnitude of MR effects ${\left(K_{\|}\right)}_{SD}$ and ${\left(K_{⊥}\right)}_{SD}$ decrease monotonically with $\phi_{p}$, as shown in Figure 8 and Figure 9. The longitudinal MR effect ${\left(K_{\|}\right)}_{SD}$ can even change sign and become negative for higher values of $\phi_{p}$ and oblate shapes; see Figure 8B,C. Similarly to magneto-induced elongation, in the case of MR effects, too, the shifting of maxima can be seen in Appendix D. For MR effects along the field direction in both cases (SCs and SDs), the optimal value of the initial aspect ratio $\gamma_{0}$ shifts toward smaller values with increasing $\phi_{p}$. On the other hand, for MR effects perpendicular to the field direction (SCs and SDs), the maxima shifts toward higher values of $\gamma_{0}$ with an increase in $\phi_{p}$.

## 5. Discussion

In the present work, we illustrated the effect of the microstructure on the mechanical properties of ellipsoidal magnetoactive elastomers. By extending the previous approach [53] to describe the distribution of magnetic particles, a much simplified analytical expression is derived depicting the chain-like and plane-like microstructures as smeared columns and disks, respectively. The proposed expression for $f_{micro}$ reproduces accurate results [31,41,63] for an isotropic particle distribution, $f_{micro}=0$, for a chain-like microstructure, $f_{micro}>0$, and a plane-like microstructure, $f_{micro}<0$. The formalism presented in Equation (Equation 9), where the shape factor $f_{macro}\neq 0$ and the microstructure description $f_{micro}\neq 0$, allows us to simultaneously study the effect of the initial shape of an MAE sample and the initial particle distribution. The optimum values of the volume fraction of particles inside a smeared structure $\phi_{p}$, where the MR effect is maximal, are obtained as a function of the initial aspect ratio $\gamma_{0}$ and the total volume fraction of magnetic particles ϕ. The effect of the microstructure shows an increase in the field-dependent modulus in the case of SCs. Yet, it is a small enhancement compared to isotropic MAEs and the enhancement reported in experimental studies [65].

The critical value of $\phi_{p}$ seen in Figure 6 directly points toward the overall increase in the effective elastic modulus of an MAE sample due to the consideration of smeared columns. According to [66,67,68], the application of an external magnetic field leads to restructuring of the particle arrangement in MAEs. Thus, one can consider the particle microstructure starting from the isotropic distribution, which changes to form smeared columns in the presence of an external magnetic field. The formation of smeared columns highly depends on the strength and orientation of an applied magnetic field as well as on the initial shear modulus of the elastomer matrix [69]. In that case, our model predicts a very high MR effect (14 fold), as depicted in Figure 10, by assuming the formation of smeared columns. It shows the large enhancement of the elastic modulus, where a major contribution arises from the elastic free energy density in addition to the field-induced stiffening. In this MR effect, the hydrodynamic reinforcement factor *k* plays a key role. The factor *k* diverges at $\phi_{p}=0.5$, at which the drastic increase in the MR effect is realized, as shown in Figure 10. The divergence of *k* is exactly equivalent to the percolation threshold defined in Ref. [66]. The analysis presented in this work provides an approximate but promising hypothesis to understand the reasoning behind the huge (over several orders of magnitude) MR effects seen in experimental studies [70]. In conclusion, the present work covers the entire spectrum of MAEs ranging from chain-like to plane-like microstructure, including the isotropic particle distribution. The proposed model shows the ability to predict the uniaxial magneto-mechanical behavior of MAEs with remarkable consistency between different microstructures.

## Figures and Tables

**Figure 1 materials-15-00645-f001:**
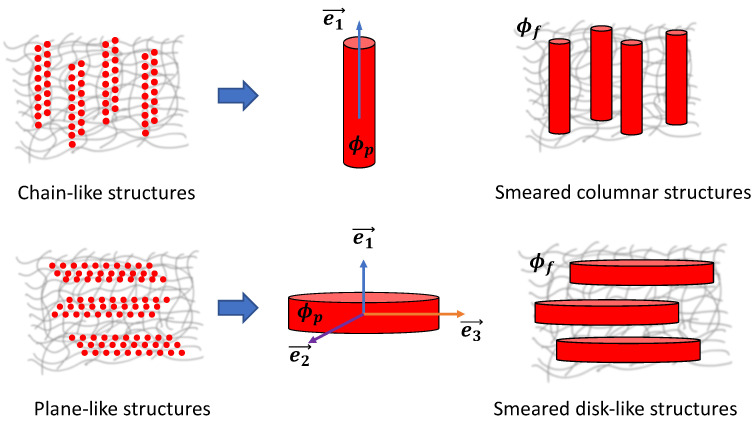
Smearing of magnetic particles (total volume fraction ϕ) into columnar and disk-like structures. $\phi_{p}$ is the volume fraction of magnetizable particles inside a smeared structure, and $\phi_{f}=\frac{\phi }{\phi_{p}}$ represents the volume fraction of smeared structures inside an elastomer matrix. MAEs, in both cases, exhibit transverse isotropy along a unit vector ${\vec{e}}_{1}$.

**Figure 2 materials-15-00645-f002:**
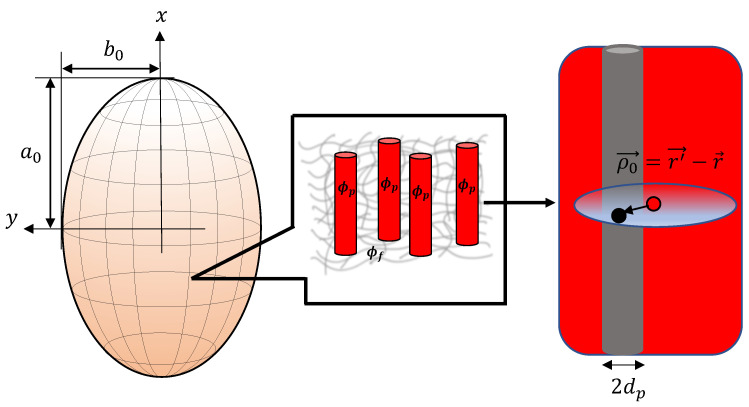
The image on the left represents schematics of an MAE sample with the shape of an ellipsoid of revolution having two equal semi-axes $b_{0}=c_{0}$ and one distinct semi-axis $a_{0}$. The magnified image on the right side depicts a smeared columnar structure.

**Figure 3 materials-15-00645-f003:**
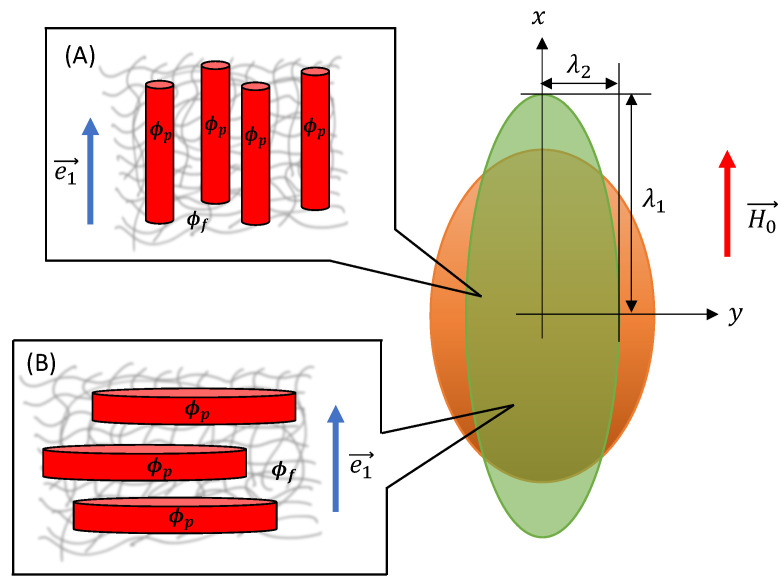
Uniaxial deformation of an ellipsoidal MAE sample with different microstructures. The orange-coloured MAE sample represents the reference configuration, while the green-coloured sample depicts the deformed configuration. $\phi_{p}$ is the volume fraction of magnetizable particles inside a smeared structure, and $\phi_{f}=\frac{\phi }{\phi_{p}}$ represents the volume fraction of smeared structures inside an elastomer matrix. (**A**) SC microstructure and (**B**) SD microstructure.

**Figure 4 materials-15-00645-f004:**
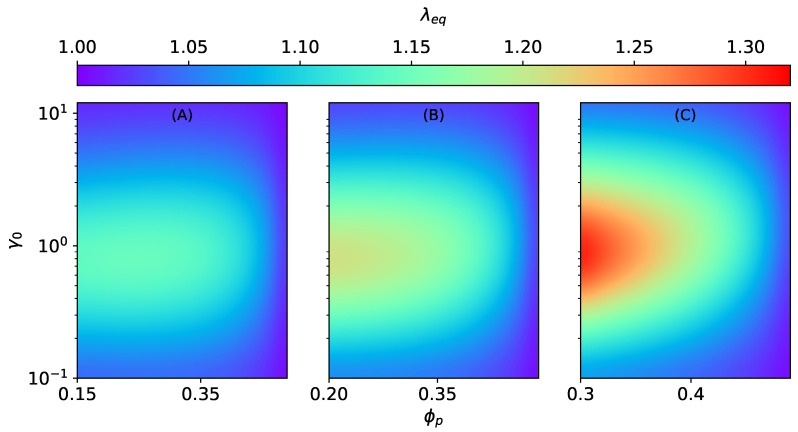
The magneto-induced elongation $\lambda_{eq}$ for MAEs with SC microstructure as a function of the initial aspect ratio $\gamma_{0}$ and the volume fraction $\phi_{p}$ at constant $|{\vec{H}}_{0}|=470$ kA/m, $G_{m}=17$ kPa. (**A**) ϕ=0.15, (**B**) ϕ=0.2, (**C**) ϕ=0.3.

**Figure 5 materials-15-00645-f005:**
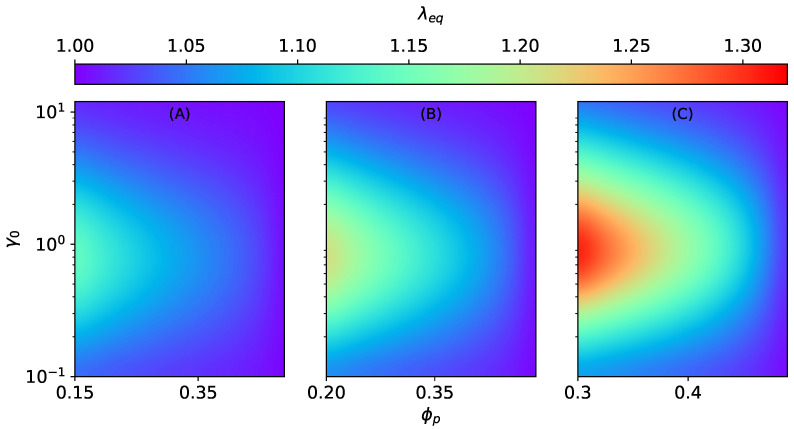
The magneto-induced elongation $\lambda_{eq}$ for MAEs with SD microstructure as a function of the initial aspect ratio $\gamma_{0}$ and the volume fraction $\phi_{p}$ at constant $|{\vec{H}}_{0}|=470$ kA/m, $G_{m}=17$ kPa. (**A**) ϕ=0.15, (**B**) ϕ=0.2, (**C**) ϕ=0.3.

**Figure 6 materials-15-00645-f006:**
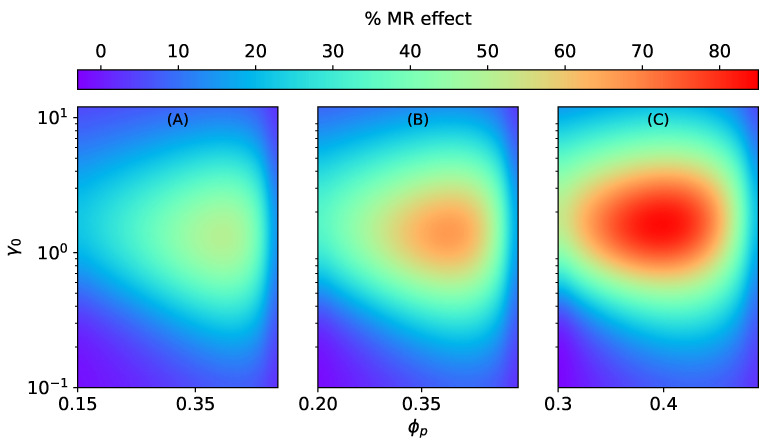
The magneto-rheological effect ${\left(K_{\|}\right)}_{SC}$ of an ellipsoidal MAE with SC microstructure stretched along the field direction at constant $|{\vec{H}}_{0}|$ = 470 kA/m, $G_{m}=17$ kPa. (**A**) ϕ=0.15, (**B**) ϕ=0.2, (**C**) ϕ=0.3.

**Figure 7 materials-15-00645-f007:**
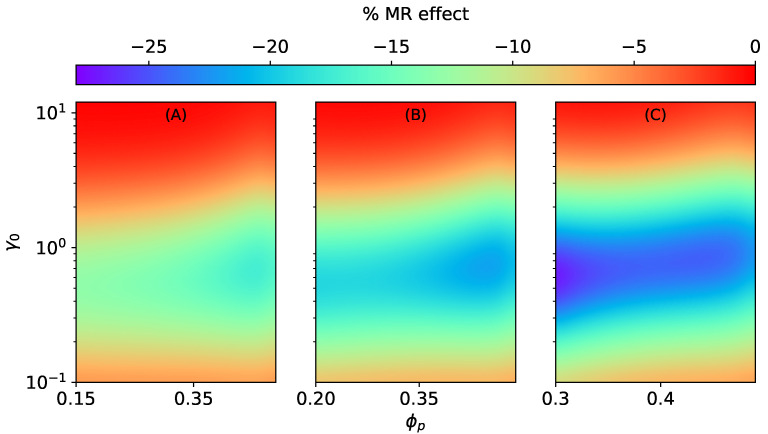
The magneto-rheological effect ${\left(K_{⊥}\right)}_{SC}$ of an ellipsoidal MAE with an SC microstructure stretched perpendicular to the field direction at constant $|{\vec{H}}_{0}|$ = 470 kA/m, $G_{m}=17$ kPa. (**A**) ϕ=0.15, (**B**) ϕ=0.2, (**C**) ϕ=0.3.

**Figure 8 materials-15-00645-f008:**
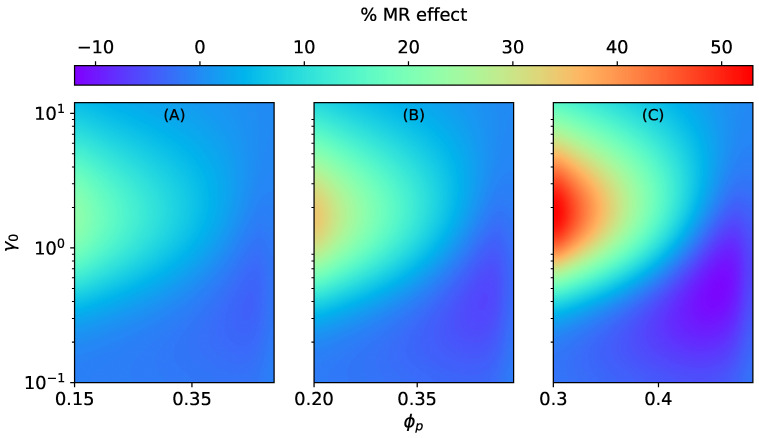
The magneto-rheological effect ${\left(K_{\|}\right)}_{SD}$ of an ellipsoidal MAE with SD microstructure stretched along the field direction at constant $|{\vec{H}}_{0}|$ = 470 kA/m, $G_{m}=17$ kPa. (**A**) ϕ=0.15, (**B**) ϕ=0.2, (**C**) ϕ=0.3.

**Figure 9 materials-15-00645-f009:**
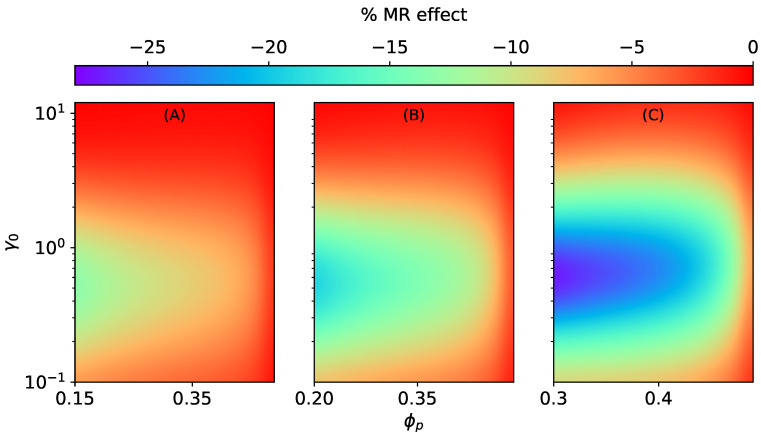
The magneto-rheological effect ${\left(K_{⊥}\right)}_{SD}$ of an ellipsoidal MAE with SD microstructure stretched perpendicular to the field direction at constant $|{\vec{H}}_{0}|$ = 470 kA/m, $G_{m}=17$ kPa. (**A**) ϕ=0.15, (**B**) ϕ=0.2, (**C**) ϕ=0.3.

**Figure 10 materials-15-00645-f010:**
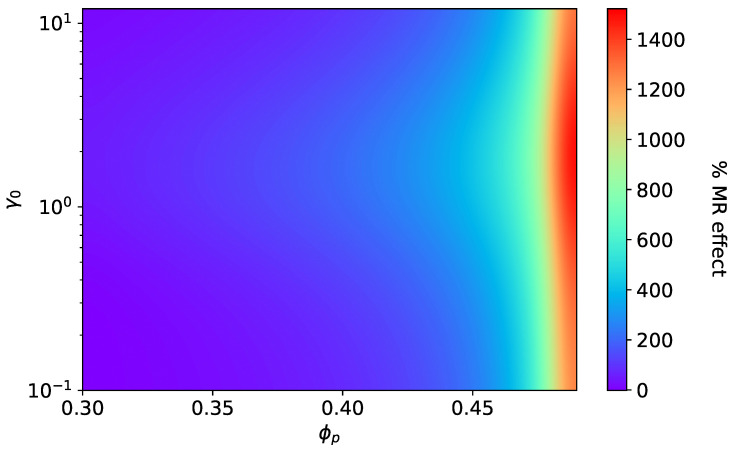
The magneto-rheological effect: Transition of isotropic MAE to anisotropic MAE with smeared columns at $|{\vec{H}}_{0}|$ = 470 kA/m, $G_{m}=17$ kPa, ϕ=0.3.

**Table 1 materials-15-00645-t001:** The values of parameters used in the numerical calculations.

Parameter	Description	Value
$G_{m}$	Shear modulus of a matrix	17 kPa
$H_{0}$	External magnetic field	470 kA/m
ϕ	Total volume fraction of magnetic particles	0.15, 0.2, 0.3
χ	Magnetic susceptibility	1000

## Data Availability

On inquiry, the data presented in this study is available from the authors.

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
