# Peer review of "Magneto-Mechanical Enhancement of Elastic Moduli in Magnetoactive Elastomers with Anisotropic Microstructures"

_materials, 2022, doi:10.3390/ma15020645_

Round 1
Reviewer 1 Report
A variety of theoretical works can be found in the literature that investigate the effect of microstructure on the mechanical properties of magnetoactive elastomers MAEs. But some limitations appear due to the consideration of perfectly ordered magnetic particules positions (chain-like and plane-like structures). Instead of discrete particle distribution, the particle positions can be smeared over an elongated columnar-like microstructure. The present work extends the formalism of chain-like structure (smeared columns, SC) to plane-like structure (smeared disk-like, SD) and attempts to simplify the formalism even further by converting the integral that describes the smeared particle microstructure in a previous work into a simple analytical expression. Explicit expressions of free energy density are provided for the SC and SD case.
Additional contribution due to anisotropic structures to the elastic free energy (transverse symmetry case) of MAEs are considered. an ellipsoidal MAE sample of two equal semi-axes and one distinct semi-axis is considered. It is studied the effect of different particle microstructures and the initial shape of an MAE sample on its mechanical properties. The magnetic particles are considered as point-like dipoles, and the linear magnetization regime is assumed with an homogenous external magnetic field aligned along the symmetry axis. Strains are restricted to uniaxial deformations.
This work can be published as it is.
Typing errors
p.20 Duplicate title in ref. 7
Reviewer 2 Report
Dear Authors,
Magneto active elastomer composite is an emerging field of research. Although lot of studies have already been done on rheological properties by different methods but still there is no suitable standard because of the advanced studies. This article represents well the development of a new model that can be useful to understand the magnetorheological behaviors of magneto active elastomer composites. However, after reading, I feel some minor revision should be made to improve the quality of this paper.
Please cite this paper with the connection of mechanical anisotropy in MSE with and without external magnetic fields in the introduction (Ref. Iron particle and anisotropic effects on mechanical properties of magneto-sensitive elastomers https://doi.org/10.1016/j.jmmm.2017.05.049).
Author assumed the particle arrangement in anisotropic MSE as smeared microstructure however the formation of such microstructure is highly depend on matrix viscosity and orientation magnetic field (Ref. Anisotropic magnetorheological elastomers with carbonyl iron particles in natural rubber and acrylonitrile butadiene rubber: A comparative study https://doi.org/10.1177/1045389X20986995). It should be better to include those factors that affect the formation of such microstructure.
Reviewer 3 Report
In this manuscript, Chougale et al. developed a finite strain mathematical model for magneto-active elastomers (MAEs). In developing theoretical model for MAEs at large strains, one of the key challenges is to formulate energy functions that can be derived directly by taking micro-structural information of the underlying magnetic particles into account. With respect to the challenge, the current manuscript develops a micro-mechanical magnetic energy function. Afterwards, they demonstrate couple of numerical examples to quantify the sophistication of their model. The argumentation of the manuscript is clear and is built upon previous works of the authors. Moreover, the mathematical modelling of MAEs is an active field of current research due to a wide range of exciting applications of the smart material. Hence, the manuscript can be published in Materials. However, it requires significant revisions. In this case, following points are to addressed:
- I found inconsistency throughout the manuscript regarding the magnetic particles shape. Sometimes, it is assumed as ellipsoidal particles (see Line 114), in other places, it is spherical (Line 155). Please be consistent !
- It is nice that the total energy function is decomposed into isotropic, anisotropic (both mechanical) and a magnetic part in which the last one is micromechanically derived. However, what about the first two energy functions (iso and ansio)? They can be also derived from micromechanics of polymer chains ( or at least authors cite relevant papers here). Furthermore, for the isotropic energy function, it is considered as the Neo-Hooke one while it is valid only for relatively small strains. I would prefer putting relevant references in which bunch of strain energies are reviewed.
- I do not understand Eqn 27 in which dJa/dF is considered. How Ja, being a parameter, is dependent on the deformation gradient F?
- Throughout the manuscript, couple of numerical simulations are performed. For that, a wide range of material parameters are used. Can they (material parameters) be related to some experimental data? or at least with relevant literature? Moreover, authors can make a table putting all relevant parameters.
- It is very weird to claim that "Tensile mechanical tests are commonly used to characterize soft elastomers" (Line 287). Please put related references with experimental studies.
- One of the main criticisms of the current manuscript is that it does not contain a single sentence or references mentioning the real applications of MAEs or why we need to work with MAEs. Please note that potential applications of MAEs are exponentially growing in recent years. Couple of good review papers nicely summarise relevant applications, e.g. https://www.sciencedirect.com/science/article/pii/S2352940721003693, https://www.sciencedirect.com/science/article/pii/S0264127521007279, https://iopscience.iop.org/article/10.1088/2399-7532/abcb0c
Once, authors agree to revise accordingly taking all my suggestions/criticisms including all relevant & important references, I will be happy to accept it.
Round 2
Reviewer 3 Report
It can be accepted now